# Efficient Gun Detection in Real-World Videos: Challenges and Solutions

## Abstract

Object detection in videos is a crucial task in the computer vision domain. Existing methods have explored different approaches to detect objects and classify the videos. However, detecting tiny objects (e.g., gun) in videos has always been a challenging and rigorous task. Moreover, the existing video analysis (detection and classification) models may not achieve high accuracy for gun detection in videos in real-world scenarios due to the lack of a large amount of labeled data. Thus, it is imperative to develop an efficient method to capture the features of tiny objects and train models that can perform accurate gun detection. To address this challenge, we make three contributions. First, we perform an empirical study of several existing video classification methods to identify the presence of guns in videos. Our extensive analysis shows that these methods may not achieve high accuracy in detecting guns in videos. Second, we propose a novel gun detection method with image-augmented training and evaluate the technique in real-world settings with different evaluation metrics. Third, our experimental results demonstrate that our proposed domain-specific method can achieve significant performance improvements in real-world settings compared to the other popular methods. We also discuss emerging challenges and critical aspects of detecting tiny objects, e.g., guns, using existing computer vision techniques, their limitations, and future research opportunities.

## 1 Introduction

Public safety is crucial to ensure the well-being and dignity of every citizen. It is highly essential to protect individuals from threats such as gun violence. Gun violence is one of the major attacks against public security. The manual monitoring systems that mainly rely on a human operator for analyzing CCTV footage to detect guns might lead to human error (Goldmeier, 2023). Moreover, several other issues, e.g., boredom, voyeurism, and biased profiling (Surette, 2005), might deteriorate the potential shooting detection scenarios. Detecting guns in videos has been a highly important task in the computer vision domain due to several factors, e.g., low video quality, small size of gun objects, and lack of large datasets to train models effectively. Identifying a tiny object (e.g., gun) from a sequence of frames is complex due to several reasons, e.g., varying lighting conditions, similarity to other objects, and low video quality (Wu et al., 2021; Leszczuk et al., 2024). Despite significant efforts, existing methods often struggle with detection accuracy, robustness, and generalization performance, especially in real-world settings. The shortcomings of the current approaches underscore the requirement for a more accurate solution that is capable of overcoming the limitations of current techniques in detecting guns in videos and performs well in dynamic environments.

To mitigate the aforementioned issues regarding human-monitored CCTV footage analysis, it is crucial to leverage Artificial Intelligence (AI) techniques to identify the guns in videos. Previous studies have explored various AI-driven approaches to tackle this problem. Most of the existing video analysis models (e.g., (Tran et al., 2018; Tong et al., 2022; Kondratyuk et al., 2021; Arnab et al., 2021; Fan et al., 2021)) are intended for action recognition in videos and have been evaluated on benchmark action recognition datasets, which might not be able to perform effective gun detection. Among the detection-oriented models, CNN (Bhatti et al., 2021; Manikandan & Rahamathunnisa, 2022; Elmir et al., 2019; Khin & Htaik, 2024), R-CNN (Olmos et al., 2018), faster R-CNN (Verma & Dhillon, 2017; Alaqil et al., 2020), and different versions of YOLO (Warsi et al., 2019; Khin & Htaik, 2024) are widely utilized. Deployment of hybrid models, e.g., Faster-RCNN-VGG16 (Olmos et al., 2018),

MobileNetV3-SSD (Ghazal et al., 2020), ResNet-YOLO (Salido et al., 2021), are also gaining attention for gun detection in videos. Despite the advancements in weapon detection technologies, current methods may be inadequate in accurately identifying guns, particularly in scenarios involving low-quality video footage and unpredictable real-world conditions. In this paper, in order to address these problems, we propose a novel accurate model that is tailored towards gun detection in videos. To this end, we also conduct a comprehensive empirical analysis of existing solutions. The key contributions of this work are as follows.

- We perform an empirical study on several video classification approaches for detecting guns in videos, including 3D CNN approach (Tran et al., 2018), pre-trained video models (Kondratyuk et al., 2021), pre-trained Transformer-based models (Tong et al., 2022), and hybrid architectures. In our empirical analysis, we underscore that none of these methods might achieve high performance in detecting guns from videos as they do for action recognition.

- We propose a novel gun detection methodology that involves a two-stage training process: (1) image-augmented training, wherein we utilize gun image datasets to introduce subtle gun features into the image model, facilitating the extraction of spatial representations from video frames that contain guns, and (2) temporal modeling, in which we incorporate a sequence model in addition to the image model to effectively capture sequential dependencies and temporal information of the video data.

- The empirical results illustrate that our proposed method outperforms state-of-the-art approaches, yielding significant improvements in performance on a specially crafted synthetic firearms action recognition video dataset.

- Our analysis investigates multiple application scenarios, drawing from real-world contexts, to highlight the imperative for developing high-performing AI-driven methods for identifying guns in video data.

- We also highlight several key challenges and limitations of existing gun detection techniques and discuss potential future research directions.

## 2 MOTIVATION

The rapid increase in crime rate in every corner of the world is alarming to human civilization. Shooting and gun violence are causing fatalities and injuries. Alone in 2018, there were more than 27 gun violence incidents that happened in US FBI (2018). In 2023, nearly 47K deaths (Hopkins, 2024) had been caused by 656 mass shootings (Archive, 2023) incidents in public places and still witnessing similar incidents almost every year in the last decade in the U.S BBC (2016); CNN (2017); CPR (2024); CNN (2022). Such violence leaves a long-term consequence for individuals, communities, and society NIHCM (2024). Table 1 shows several examples of gun violence incidents in public/crowded places, with the fatalities in the US in the last few years. The first White House Office of gun Violence Prevention was established in September 2023 "to reduce gun vio-

Table 1: Example of Recent Gun Violence Fatalities in the US

| Year | Location | Fatalities , Injuries |
|------|----------|-----------------------|
| Sept. 2024 | Georgia (at school) | 2, 2 (CNN, 2024a) |
| Jul. 2023 | Illinois (at parade) | 7, 83 (ABC, 2022) |
| May, 2022 | Texas (school) | 21, 17 (Tribune, 2024) |
| Mar. 2021 | Colorado (superstore) | 10, 2 (CPR, 2024) |
| Aug. 2019 | Ohio (Bar) | 9, 27 (CNN, 2019) |
| Oct. 2017 | Nevada (Festival) | 59, 851 (NBC, 2017) |

lence, which has ravaged communities across the country, and implement and expand upon key executive and legislative action which has been taken to save lives" WH (2023). Further, several other challenges motivated us to engage in this research. We briefly discuss them as follows.

### 2.1 CHALLENGES OF HUMAN MONITORED SURVEILLANCE SYSTEMS

The proliferation of Closed-Circuit Television (CCTV) surveillance cameras in public/crowded places and critical areas has significantly enhanced global security infrastructure, providing the ability to prevent, detect, and respond to potential threats. It consists of one or more cameras connected

to one or more monitors. Human operators closely monitor the video footage and take necessary actions for any unusual events, including gun violence. Therefore, the efficacy of such systems is highly dependent on the human judgment capability to identify any unusual events or threats among high volumes of footage, specifically when it might take just a few seconds. However, manually reviewing and analyzing these videos is a daunting task and may lead to human errors at late hours and sometimes emotional trauma. On top of that, once the violence is identified, then the human identifier has to report the incident to another end to activate alarms and call the security forces. This entire process sometimes might take a long time, and by then, such a violent incident might cause huge fatalities. There are several recent incidents where, despite having CCTV-enabled human supervision, gun violence took place and had a massive impact (BBC, 2016; CNN, 2024b). It is highly necessary to develop and integrate an automated AI-based high-performing model to detect gun violence in videos.

## 2.2 CRITICAL ASPECTS OF AUTOMATED GUN VIOLENCE DETECTION MODELS IN VIDEOS

Building a gun detection model for videos is itself a very critical task for several reasons. First, usually, the gun appears in videos as a very tiny object and there can be numerous gun categories with different sizes and shapes, which makes it difficult to develop a universal detection model. Second, the video quality from where the guns are to be detected is one of the major dependencies behind the performance of the model (O'Byrne et al., 2022). Videos captured under various lighting conditions/effects, i.e., shadows, and reflections can deteriorate (Leszczuk et al., 2024) the performance of a model, even if it performs well on the standard

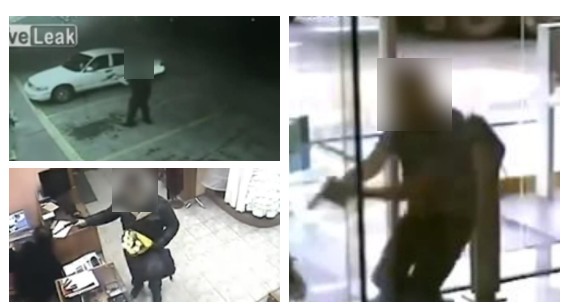

Figure 1: Video frames of CCTV footage with Guns from a real-world dataset

quality of videos. Third, guns may be occluded fully or partially in videos, or obscured by other objects. Also, other similar gun-shaped objects (e.g., camera tripods) might be held at unusual angles in real-world incidents, making identification even harder and leading to reduced accuracy, increased false positives, and challenges in feature extraction (Aqqa et al., 2019). We show some sample video frames with guns from the UCF crime dataset (Sultani et al., 2018) in Figure 1. From the illustration, we can observe that in a real-world scenario, the gun appears as a very tiny object with other similar objects in the frame, and the video quality might be very low.

## 2.3 LACK OF DOMAIN-SPECIFIC HIGH-PERFORMANCE MODEL AND DATASET

Previously proposed approaches, e.g., (Kondratyuk et al., 2021; Hara et al., 2017; Karpathy et al., 2014; Tong et al., 2022; Yu & Li, 2017) were mostly focused on action recognition such as walking, playing, jumping in videos. Moreover, for evaluating these models' literature used different action recognition datasets or various sports classification datasets, such as the UCF101 action recognition dataset (Soomro, 2012), The Kinetics Human Action Video Dataset (Kay et al., 2017), HMDB51 (Kuehne et al., 2011), Sports-1M (Karpathy et al., 2014). So, there is a significant lack of proper gun violence recognition datasets in the literature. Consequently, a substantial research gap persists in developing an efficient and high-performing gun violence recognition model, which underscores the need for further investigation in this critical area.

To combat the above challenges, first, we perform an empirical study to evaluate the performance of current approaches of video classification to detect guns in videos on a synthetic gun action recognition dataset (Ruiz Santaquiteria et al., 2024) and illustrate the need for a high-performing model for such tasks. Then, we propose and evaluate a new model that is specifically focused on gun violence identification on both crafted gun action recognition dataset (Ruiz Santaquiteria et al., 2024) and real-world UCF crime dataset (Sultani et al., 2018) by considering the critical challenges.

## 3 PROBLEM STATEMENT

In the real world, machine learning (ML) models learn through a vast amount of data in the training phase for different learning tasks. For instance, the state-of-the-art high-performing image classifi-

cation models such as ResNet (He et al., 2016), EfficientNet (Tan, 2019), are trained over ImageNet (Deng et al., 2009) which contains more than 1M labeled samples of 1000 categories. Moreover, using (fine-tuning) those models in another domain for any learning task also requires a significant amount of domain-specific labeled data for achieving competitive performance. As discussed, the existing video models are more focused on action recognition (Kondratyuk et al., 2021; Tran et al., 2018; 2015; Ji et al., 2012). Also, the advanced transformer-based pre-trained video recognition models, e.g., VideoMAE (Tong et al., 2022), ViViT (Arnab et al., 2021), Slowfast (Feichtenhofer et al., 2019), X3D (Feichtenhofer, 2020), MVit family (Fan et al., 2021; Li et al., 2022), which have been trained on labeled action recognition video datasets, thus mostly suitable for action recognition tasks in videos.

In this paper, we consider gun detection in videos as a classification problem. In order to achieve high performance in this domain, we require labeled datasets with a significant amount of samples to train a new model or fine-tune the existing models. There is a huge scarcity of labeled video datasets for such tasks in this specific domain. Therefore, current methods may not achieve high performance either in training or fine-tuning due to the lack of training samples. We illustrate the performance of several existing representative video classification methods and discuss it in detail in the experiment section (Section 5).

**Transfer Learning** is a widely used technique that benefits from a pre-trained model and reuses the existing model's knowledge for a similar task (Zhuang et al., 2020). In order to mitigate the above-mentioned issues, transfer learning can be utilized to perform gun detection by leveraging the pre-trained weights of existing models for a better understanding of the features of guns in videos. The fundamental idea behind transfer learning is to use the knowledge learned from one task (the *source* task) of any domain and apply it to another related task (the *target* task) to the same or different domain (Zhuang et al., 2020; Pan & Yang, 2009). Formally, a domain $D$ contains a feature space $X$, ($X$ is a set of instances defined as $\{x_1, x_2, ...., x_n\}$) and a probability distribution $P(X)$. For a specific domain, $D = \{X, P(X)\}$. A task $\tau$ is defined as $\tau = \{Y, f\}$, where $Y$ denotes a label space defined as $\{y_1, y_2, ...., y_n\}$ and $f$ is a prediction function, $f : X \rightarrow Y$, which is used to predict the corresponding label $f(x)$ of a new instance $x$. The task $\tau$ is defined as $\tau = \{Y, f(x)\}$, which is learned from the training instances consisting of pairs $\{x_i, y_i\}$, where $x_i \in X$, and $y_i \in Y$. Given a source domain $D_s$ and source learning task $\tau_s$ and a target domain $D_t$ and target learning task $\tau_t$ (e.g., gun detection), where $D_s \neq D_t$ and $\tau_s \neq \tau_t$, we aim to identify effective transfer learning strategies that can improve the learning of the target prediction function, $f_t$ in $d_t$ utilizing the knowledge in $D_s$ and $\tau_s$.

## 4 VIDEO CLASSIFICATION METHODS

### 4.1 EXISTING VIDEO CLASSIFICATION METHODS

The video classification problem has been explored using different approaches in the literature. In this paper, we explore several representative video classification models from the existing studies. Here, we describe them briefly.

#### 4.1.1 3D-CONVOLUTONAL NEURAL NETWORK (3D-CNN)

3D-CNN was proposed in order to effectively capture both spatial and temporal dimensions of video data (Ji et al., 2012). The fundamental idea was to build a 3D architecture as a 3D convolutional feature extractor. Tran et al. (2015) proposed Convolutional 3D (C3D) which is basically a 3D kernel size of $T \times K \times K$ (where $T$ is temporal depth, and $K$ is spatial kernel size) that slides over the entire volume of video in three dimensions to extract spatiotemporal information on large supervised training for different video analysis tasks. Another approach to implementing 3D CNN for video analysis is factorizing 3D kernel into 2D spatial convolution and 1D temporal convolutions introduced by Tran et al. (2018). They proposed an R(2+1)D convolutional block within residual architecture, which better helps to understand the spatiotemporal features more effectively than pure 3D-CNN by increasing the nonlinearity between these two operations.

#### 4.1.2 MOVINET

Mobile video networks (MoViNets) is a family of 3D-CNN video recognition models which is memory and computation-efficient (Kondratyuk et al., 2021), optimized for resource-constrained

devices. Unlike vanilla 3D-CNN, it doesn't require high computational resources for training and inference. Essentially, it contains three major components. First, **MoViNet search space,** which is built based on MobileNetV3 (Howard et al., 2019) and inspired by X3D (Feichtenhofer, 2020), the 2D blocks of MobileNetV3 are expanded to handle the 3D video input. It creates a neural architecture search (NAS) which obtains an optimal neural architecture configuration to trade off the performance and efficiency based on the specific use cases. Second, in order to deal with the memory budget, instead of processing the entire video at once, it splits the entire video into smaller $n$ sub-clips, called **Stream-buffer.** It mitigates the issues of recomputing the frame activations due to overlapping and preserves long-range dependencies by caching feature activation at each edge of sub-clips. Once it iterates over $n$ sub-clips, the whole feature map is obtained by concatenating the activations cached at each $i^{th} < n$ step. Third, **temporal ensembles** allows the creation of two identical models separately by halving the frame rate and offsetting by one frame. Finally, before the softmax, the arithmetic mean is performed on unweighted logits. Thus, it ensembles two models with the same computational cost, leading to enriched predictions. Several versions (pre-trained on Kinetics-600 (Kay et al., 2017)) are available depending on the capacity of the users' device.

### 4.1.3 VIDEOMAE

Inspired by the success of masked autoencoding in images (He et al., 2022; Bao et al., 2021) and NLP (Devlin, 2018), authors in Tong et al. (2022) proposed video masked autoencoder (VideoMAE). Video MAE is a self-supervised learning technique designed specifically for video pre-training. It overcomes the requirements of large-scale labeled datasets and introduces a vanilla vision transformer on the video dataset without using pre-trained models or extra data. During the pre-training phase, it randomly masks 90-95% of video frames using the **tube masking** technique due to the temporal redundancy, and it yields great efficiency in terms of computational cost due to the asymmetric encode-decode architecture. It successfully trains the basic ViT backbone (Dosovitskiy, 2020) with relatively small video datasets, e.g., SSV2 (Goyal et al., 2017), UCF101 (Soomro, 2012), HMDB51 (Kuehne et al., 2011). The model's task is to reconstruct the masked pixels of the downsampled video clips from the unmasked ones and this reconstruction process allows it to effectively learn features and patterns (both spatial and temporal) in the videos. The pre-trained model can further be fine-tuned with smaller labeled datasets for any downstream tasks, classification, or detection.

### 4.1.4 HYBRID ARCHITECTURES

Hybrid architectures are also gaining popularity for different video analysis tasks, e.g., event classification in videos (Zhang & Xiang, 2020), facial expression classification (Abdullah et al., 2020), gesture recognition (Hu et al., 2018). The key idea of this twofold architecture is extracting the spatial features from the video frames with deep neural network (DNN) models (e.g., VGG) and handling the temporal features with sequence models (e.g., LSTM, Transformer). Different combinations of these hybrid models are able to achieve high performance for action recognition and earlier-mentioned tasks as state-of-the-art video analysis models on benchmark datasets (Paul, 2021), (Mahmoud, 2024).

Though these advancements have shown their potential towards video classification area, these methods are primarily focused on action recognition with various datasets such as UCF-101 action recognition dataset (Soomro, 2012), Kinetics Human Action Video Dataset (Kay et al., 2017), HMDB51 (Kuehne et al., 2011), Sports-1M (Karpathy et al., 2014). Thus, these models can achieve high accuracy in classifying different action classes. However, it's unclear how these methods will perform in order to find the presence of tiny objects such as guns in videos and, more importantly, in real-world environments.

### 4.2 IMAGE AUGMENTED TRAINING FOR GUN VIDEO CLASSIFICATION

To overcome two major issues of existing methods, namely the lack of feature learning abilities to identify guns in videos and the lack of sufficient gun detection video datasets, we propose a novel gun detection method with an image-augmented trained model wrapped with a sequence model.

First, we employ several state-of-the-art image models (e.g., VGG, ResNet, and MobileNet) for image-augmented training, where the models are pre-trained over the ImageNet benchmark dataset. We apply the transfer learning technique to take advantage of the pre-trained weights/parameters $(\theta_s)$ of source task $(\tau_s)$ of existing image models, improving the model's ability to capture the features of gun images for the target task $\tau_t$, i.e., gun detection.

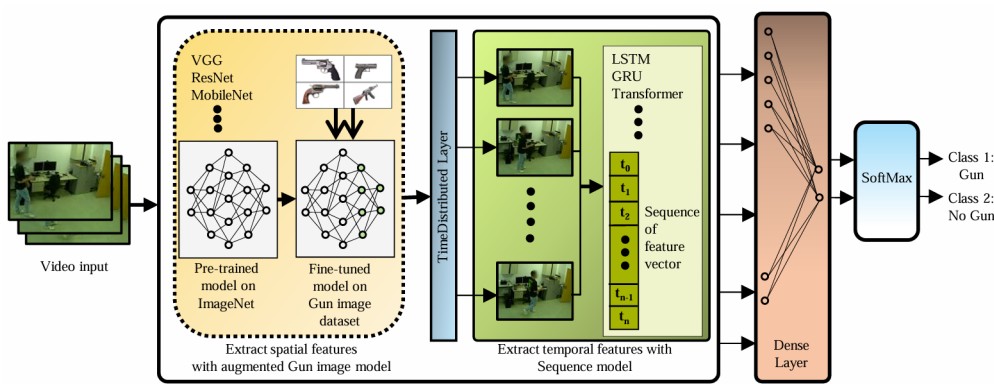

Figure 2: Building blocks of gun image-augmented model for detecting guns in videos.

Further, we apply **fine-tuning** (Sharif Razavian et al., 2014) technique within transfer learning in which the parameters trained for the source task $\tau_s$ are not fully retained but rather adjusted for the target task $\tau_t$. It gets initialized by $\theta_t = \theta_s$, where $\theta_s$ and $\theta_t$ are the model parameters for source and target tasks, respectively. Then it can be updated during training for target task $\tau_t$ as

$$\theta_t = \theta_s + \Delta\theta \tag{1}$$

Here, $\Delta\theta$ is the updates on $\theta_s$ for the target task. Finally, the optimization is done by minimizing the loss function on target task $\theta_t$. For our case, the state-of-the-art models can be considered as the source task and the gun detection tasks as the target task. We leverage the model parameters $\theta_s$ trained for $\tau_s$ and fine-tune it as $\theta_t$ as shown in Equation 1 for $\tau_t$, i.e. gun detection.

We perform fine-tuning the pre-trained models with the gun images (see details in Section 5) by augmenting it using different data augmentation techniques in computer vision, e.g., flipping, rotating, scaling, as well as color and intensity adjustments. Since these image models have learned the gun domain-specific features, we employ these models to extract the spatial features from the downsampled frames of videos that contain guns.

In order to handle the temporal features of the videos, we employ a sequence model on top of our built gun image-augmented classifier. We consider the videos as a sequence of frames, $X = \{x_1, x_2, ...., x_t\}$, $t$ is denoted as the length of the sequence. The input sequence $X$ is mapped to hidden state $h_t$ as $(x_t, h_{(t-1)}; \theta)$, where $\theta$ is model parameters. The hidden state gets updated as $(h_{(t-1)}, x_t; \theta)$, and the output is produced at each time step from the inputs of the hidden state as $(h_t, \theta)$. During the training, the loss function $L(Y, Y')$ is minimized with respect to model parameters $\theta$. Here, $Y$ and $Y'$ are denoted as ground truths labels and predicted labels, respectively (Sutskever, 2014; Goodfellow et al., 2016).

In Figure 2, we show the overview of our proposed method. An ordered sequence of frames, i.e., videos, are given as input to the gun image-augmented model, which has been fine-tuned with the gun image datasets (see Table 2, $2^{nd}$ row) for extracting the gun features from it. Then TimeDistributed layer (Qiao et al., 2018) is applied to each of the extracted frame features by the previous layer, and it creates a sequence of feature vectors for temporal correspondence. Then the sequence model deals with the temporal features and passes them to the next layer. Finally, two dense layers followed by a softmax perform the prediction.

### 4.3 VISUAL EXAMPLES OF FEATURE EXTRACTION WITH CLASS ACTIVATION MAP

Class activation map is a popular visualization technique in the computer vision domain. It provides valuable insights into the feature extraction process of DNN models through visualization of the most influential spatial regions for output prediction. In this paper, we use Gradient-weighted Class Activation Mapping (Grad-CAM) (Selvaraju et al., 2017) for visualizing the most significant region where the model extracts the spatial features of guns in the video dataset. In order to generate the Grad-CAM $L^c_{Grad-CAM}$, the output $y$ of class $c$ is computed w.r.t. feature maps $A^K$ of size $(H \times W)$ of the convolutional layer.

Here, $H, W$ are the height and width of $A^K$, respectively. The gradients are the global averaged pooled during backpropagation to obtain the neuron importance weights as Equation 2. $\alpha_K^c$ represents a partial liberalization of the network downsampled from $A^k$ and extracts the important weights for target class $c$.

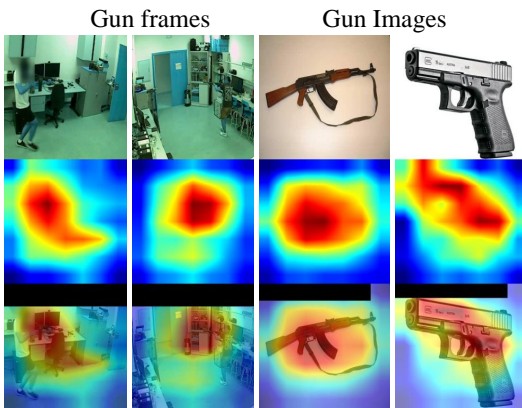

Gun frames     Gun Images

$$\alpha_K^c = \frac{1}{H \times W} \sum_{i=1}^{H} \sum_{j=1}^{W} \frac{\partial y^c}{\partial A_{i,j}^k} \qquad (2)$$

Then a weighted combination of forward activation maps is performed, followed by a $ReLU$ activation function for obtaining the $L_{Grad-CAM}^c$ as

Figure 3: The class activation heatmap for downsampled video clips and gun images

$$L_{Grad-CAM}^c = ReLU \sum_K \alpha_K^c A^K \qquad (3)$$

Figure 3 illustrates the region of spatial interest where the features of the gun class have been captured for our proposed model using the heatmap produced by Grad-Cam method (Selvaraju et al., 2017). The two columns represent the class activation map for two representative downsampled frames with gun extracted from firearm action recognition dataset (Ruiz Santaquiteria et al., 2024) and in the last two columns we show it for gun image dataset collected from (Olmos et al., 2018; Yongxiang et al., 2022).

## 5 EXPERIMENTAL ANALYSIS

### 5.1 DATASET PREPARATION AND EVALUATION METRICS

For our experiments, first, we perform empirical studies over several state-of-the-art video classification methods on the baseline action recognition dataset and the gun action recognition video dataset and compare the performance. Then we build our proposed model over image-augmented training with a sequence model and apply it to the gun action recognition dataset and UCF crime dataset. This entire experiment was conducted on a GPU server with six NVIDIA A100-PCIE (40GB memory).

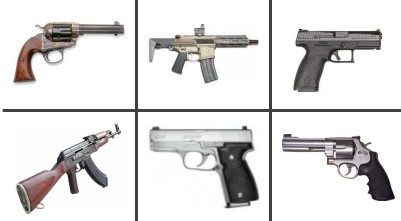

We employ several image and video datasets to build and evaluate our model. For performing gun image-augmented training on the pre-trained models, we merge two weapon

Figure 4: Sample Gun images for image-augmented training.

detection datasets (Olmos et al., 2018; Yongxiang et al., 2022) from publicly available repositories. These datasets contain different categories of guns and other handheld weapons.

For this experiment, we select different categories of over 10k gun images from the weapon detection datasets (Olmos et al., 2018; Yongxiang et al., 2022) for gun class. Figure 4 represents the sample images by which we build a gun image-augmented trained model. For the other class (which does not contain any gun), we randomly pick a similar number of images (other than gun categories) from various baseline image datasets, e.g., ImageNet (Deng et al., 2009), CIFAR-10 (Krizhevsky, 2009), which are commonly used in computer vision. In the em-

Table 2: Dataset Information

| Dataset | Samples per Class |
|---|---|
| Gun dataset for image augmented training (Olmos et al., 2018) (Yongxiang et al., 2022) | Gun: 10,381 NoGun: 10,380 |
| Firearms Action Recognition Dataset (Ruiz Santaquiteria et al., 2024) | Gun: 303 NoGun: 95 |
| UCF Crime Dataset (Sultani et al., 2018) | Gun: 50 NoGun: 50 |

pirical experiment and evaluating our proposed model, we employ a firearm recognition and detec-

tion dataset (Ruiz Santaquiteria et al., 2024) that has been specifically crafted by considering the factors of the CCTV footage of violent incidents, i.e., camera angle, lights, arm position in real-world gun violence video. Also, we employ the UCF crime dataset, which is a publicly available real-world dataset containing the shooting category. Table 2 shows information on analyzed datasets.

In order to evaluate the baseline methods, and our proposed methods, we use several widely-used evaluation metrics for classification. These metrics include Accuracy (Acc.), Precision, Recall, F1 score, Area Under the Curve (AUC), and Receiver Operator Characteristic (ROC).

## 5.2 EMPIRICAL COMPARISON OF EXISTING VIDEO CLASSIFIERS

Table 3: Performances comparison of the empirical study of the existing video models for UCF action recognition dataset (Soomro, 2012) and firearms action recognition dataset (Ruiz Santaquiteria et al., 2024)

| Existing Methods | UCF action recognition dataset (Acc.) | Firearm action recognition dataset (Acc.) |
|---|---|---|
| 3D-CNN (Tran et al., 2018; Tesnsorflow, 2017) | 75.31% | 67.39% |
| CNN-Transformer (Paul, 2021) | 89.29% | 83.92% |
| MoViNet (Kondratyuk et al., 2021; Tesnsorflow, 2021) | 97.94% | 83.33% |
| VideoMAE (Tong et al., 2022; MCG-NJU, 2022) | 93.16% | 91.34% |
| MobileNetGRU (Mahmoud, 2024) | 93.33% | 95.04% |

In order to evaluate how accurately the existing methods can classify the videos that contain guns, we perform an empirical study over several video analysis models which are primarily built focusing on classifying action classes. As discussed in Section 4, we choose five representative video analysis models for classifying gun videos in the firearm action recognition dataset Ruiz Santaquiteria et al. (2024) and compared their performance (in accuracy) as it gives for the UCF human action recognition dataset. In Table 3, we observe that 3D-CNN and CNN-Transformer do not even achieve comparable performance for firearm action recognition datasets with the UCF action recognition dataset. Where the pre-trained model MoViNet gives 97.94% accuracy on action recognition tasks, it achieves only 83.33% for classifying gun videos. Transformer-based model, VideoMAE achieves similar performance for gun video classification but is still less accurate than action recognition tasks. Finally, the MobileNetGRU (hybrid model) gives slightly better accuracy by 1.71% in classifying gun videos on the manually crafted dataset in a controlled environment, however, it might not perform well in real-world cases where several challenges persist as discussed in Section 2.

Table 4: Experimental results of the proposed model for Gun detection in videos with different configurations on Firearm action recognition dataset (Ruiz Santaquiteria et al., 2024).

| Method Configuration | Acc. | Precision | Recall | F1 | AUC |
|---|---|---|---|---|---|
| VGG + LSTM | 97% | 95% | 100% | 97% | 99.59% |
| VGG + GRU | 97% | 100% | 94% | 97% | 98.66% |
| VGG + Transformer | 97% | 96% | 98% | 97% | 99.29% |
| ResNet + LSTM | 99% | 100% | 99% | 99% | 99.92% |
| **ResNet + GRU** | **100%** | **100%** | **100%** | **100%** | **100%** |
| **ResNet +Transformer** | **100%** | **100%** | **100%** | **100%** | **100%** |
| MobileNet + LSTM | 99% | 100% | 98% | 99% | 99.66% |
| **MobileNet + GRU** | **100%** | **100%** | **100%** | **100%** | **100%** |
| **MobileNet + Transformer** | **100%** | **100%** | **100%** | **100%** | **100%** |

## 5.3 EVALUATION OF IMAGE-AUGMENTED GUN DETECTION MODEL

Our proposed gun detection model aims to correctly classify the videos that contain guns in any of its frames. To effectively extract the spatial gun features from videos, image-augmented training is performed by fine-tuning the pre-trained image classifiers, including VGG (Simonyan & Zisserman, 2014), ResNet (He et al., 2016), and MobileNet (Howard, 2017) with the gun image dataset we constructed. To deal with the temporal correspondence of the videos, we add sequence models, including LSTM (Hochreiter, 1997), GRU (Chung et al., 2014), and Transformer (Vaswani, 2017) on top of the image-augmented trained models. We present and evaluate the performance of our

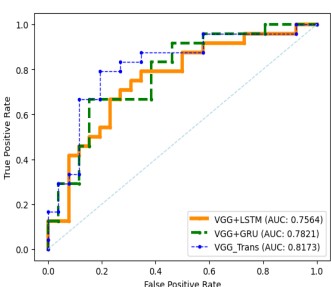 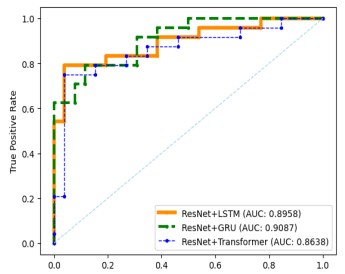 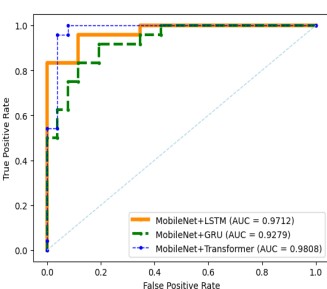

(a) VGG with different sequence models

(b) ResNet with different sequence models

(c) MobileNet model with different sequence models

Figure 5: ROC curves and AUC scores of our proposed method with different configurations for UCF crime dataset.

proposed method in Table 4 for the firearm action recognition dataset in several combinations of different image-augmented trained models with sequence models. It achieved significant performance improvements, and some variants outperform the existing methods in terms of classification accuracy. In Table 4, we show that ResNet and MobileNet combined with GRU and Transformer give a perfect 100% accuracy across all the evaluation metrics in detecting the presence of guns in videos. The values of other evaluation metrics we use in this paper also show the superiority of our methods in terms of accurately detecting guns in videos for firearm action recognition datasets. Also, the lowest performing combinations (VGG combinations with sequence models) of our proposed method perform better than the highest performing model we explored in our empirical study in Table 3 for the same dataset.

Table 5: Experimental results of the proposed model for Gun detection in videos with different configurations on UCF Crime dataset (Sultani et al., 2018).

| Method Configuration | Acc. | Precision | Recall | F1 | AUC |
|---|---|---|---|---|---|
| VGG + LSTM | 72% | 71% | 77% | 74% | 75.64% |
| VGG + GRU | 66% | 70% | 62% | 65% | 78.02% |
| VGG + Transformer | 78% | 78% | 81% | 79% | 81.73% |
| ResNet + LSTM | 80% | 83% | 77% | 80% | 89.58% |
| ResNet + GRU | 76% | 79% | 73% | 76% | 90.86% |
| ResNet +Transformer | 86% | 81% | 96% | 88% | 86.38% |
| MobileNet + LSTM | 90% | 86% | 96% | 91% | 97.11% |
| MobileNet +GRU | 84% | 88% | 81% | 84% | 92.78% |
| **MobileNet + Transformer** | **92%** | **89%** | **96%** | **93%** | **98.08%** |

As presented in table 5, we also evaluate the performance of our proposed model on the UCF crime dataset (Sultani et al., 2018), which is constructed through a more real-world environment, e.g., low-light/shadow, and poor video quality. We observe that the combination of MobileNet and Transformer achieves 92% accuracy, 89% precision, 96% recall, and 93% F1 score. It also archives 98.08% AUC score on the UCF crime dataset. In Figure 5, we plot the ROC curves and AUC scores for all the combinations of our proposed models for the UCF crime dataset. This indicates that our proposed model can also achieve robust performance in detecting guns in videos shot in low quality, poor lighting conditions, i.e., challenging real-world scenarios compared with crafted environments.

## 6 DISCUSSION

The existing models struggle to achieve high performance in detecting guns in videos due to several reasons. First, existing models are not trained specifically for identifying gun features in video frames, rather these are trained for generic action recognition tasks (Kondratyuk et al., 2021; Tong et al., 2022). Second, while techniques such as 3D-CNN (Ji et al., 2012; Tran et al., 2018), can keep correspondence between spatiotemporal features of the action recognition tasks, they can not adequately capture the spatial features of guns in sequences of frames. VideoMAE's tube masking

(Tong et al., 2022) technique might mask the portion where the gun appears in the video frames. Masking random cubes with extremely high ratios may cause the elimination of the spatial features of a gun where it appeared in the video. So, for action recognition, it is able to reconstruct the masked parts by finding the corresponding patches in adjacent frames, however, this idea might not work for effectively capturing small object features, e.g., gun. Due to these types of architectural designs, these models fail to learn the subtle gun features or features of similar small objects that appear in the video. Third, while several existing methods demonstrate comparable performance in action classification on synthetic datasets (Ruiz Santaquiteria et al., 2024), their effectiveness may significantly diminish in real-world scenarios. For instance, in the UCF crime dataset (Sultani et al., 2018), where the instances are taken from real-world shooting violence, the performance might severely deteriorate for existing models. Our proposed method addresses these issues by feeding spatial gun features to the model through image-augmented training. Thus, it is able to effectively capture the features of tiny objects, i.e., guns, in videos. Further, the sequence model handles the temporal correspondence with respect to the spatial features. According to the empirical analysis, without image-augmented training, the combination of MobileNet and GRU leads to 95.04% accuracy (i.e., last row of Table 3) for the firearm action recognition dataset. Further, the image-augmented training may achieve 100% accuracy (i.e., $9^{th}$ row, second column of Table 4) on the same dataset. Therefore, image-augmented training plays a significant role in terms of performance improvement in order to detect guns in videos. Through our experimental analysis, we demonstrated that our proposed method outperforms the state-of-the-art methods in classification accuracy and exhibits high generalization performance in real-world scenarios like the UCF crime dataset.

One potential limitation of our proposed method could be its low performance in more challenging scenarios, e.g., videos with poor quality in low light conditions and heavily rainy or stormy environments, as described in Section 2, where deep ensembles Lakshminarayanan et al. (2017); Liu et al. (2019); Wu et al. (2023) can serve as a potential solution for enhancing the overall robustness under these adverse situations.

# 7 RELATED WORK

Weapon detection in videos has been explored through various approaches in the literature. Very few of them obtained classification-based approaches, e.g., Bhatti et al. (2021) explored the siding window technique through pre-trained image classifiers. Olmos et al. (2018) proposed a pistol detection method in videos guided by the features extracted by the VGG-16 model under the sliding window technique. On the other hand, state-of-the-art detection-based approaches are also used to address this problem. R-CNN (regions with CNN features) (Girshick et al., 2014), faster R-CNN (Ren et al., 2016), different versions of YOLO (You Only Look Once) (Redmon, 2016) family, and Single Shot Detector (SSD) (Liu et al., 2016) are widely used in literature to detect guns or other handheld weapons (e.g., knives, grenades) in videos (Ahmed et al., 2022; Hashmi et al., 2021; Bhatti et al., 2021; Ingle & Kim, 2022). However, these methods may not be able to achieve robust performances specifically in more challenging real-world scenarios.

# 8 CONCLUSION

This paper addressed the crucial challenges of gun detection in videos by contributing to three main directions. Through an empirical analysis of existing video classification methods, we identified their drawbacks in terms of gun detection accuracy in videos. We then leveraged this empirical analysis and developed a novel gun detection model utilizing an image-augmented training technique, which was evaluated using a wide range of datasets, including the synthetic gun action recognition dataset and real-world UCF crime dataset. Our empirical results verify that our approach not only outperforms state-of-the-art methods in terms of classification accuracy but also demonstrates robust performance in real-world scenarios. Further, we highlighted other challenges associated with detecting small objects in videos, such as guns, showcasing the limitations of current computer vision techniques. Our findings pave the way for future research opportunities to enhance object detection capabilities, ultimately contributing to improved gun detection accuracy.

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
