# OpenReview forum: "Efficient Gun Detection in Real-World Videos: Challenges and Solutions"
_ICLR.cc/2025/Conference — Submitted to ICLR 2025_

### Official Review · Reviewer_mKKk · 2024-10-25

**Soundness:** 2
**Presentation:** 2
**Contribution:** 2
**Rating:** 3
**Confidence:** 4

**Summary:**

This paper aims to address an important and interesting gun detection tasks. The authors evaluate several video classification methods for detecting guns in videos and summarize that existing approaches cannot work well to handle the gun detection task. The authors propose a novel two-stage training methodology combining image-augmented training (introduce subtle gun features into the model) and temporal modeling (capture sequential dependencies), resulting in significant performance improvements on a synthetic firearms action recognition dataset. The analysis in this paper also emphasizes the need for advanced AI-driven gun detection methods in video data, highlights current challenges and limitations of existing techniques, and suggests directions for future research.

**Strengths:**

1) The research topic for gun detection from videos is interesting and valuable.

2) The authors did an empirical study of existing video classification methods to detect the presence of guns in videos.

3) The authors also investigate the challenges of real-world gun video detection.

**Weaknesses:**

1) Very out-of-dated methods and comparisons. The authors should compare some recent video classification and detection algorithms.

2) limited novelty, this paper just combines existing algorithms together, such as Video MAE and different network backbones (LSTM and Transformer). Very old backbones and baselines cannot illustrate the superior performance of the existing work.

3) The motivation part is too redundant. The authors should shorten this part.

4) It is not convinced to include Transfer Learning in this paper since this paper did not perform corresponding experiments.

**Questions:**

It is not clear the relationship between the video activity recognition and gun detection from the videos in this paper. The authors should focus on the gun detection from the whole videos and propose to construct the gun detection dataset and benchmark.

---

### Official Review · Reviewer_YaTr · 2024-10-27

**Soundness:** 2
**Presentation:** 2
**Contribution:** 1
**Rating:** 3
**Confidence:** 4

**Summary:**

In this paper, the authors propose a gun detection method in real-world videos. First, they perform an empirical study of several existing video classification methods to identify the presence of guns in videos. Then, they use transfer learning to extract frame-level features followed by a sequential model for video classification. Third, they conduct experiments on the Firearm action recognition dataset and UCF Crime dataset to show the performance.

**Strengths:**

1. The method is clear and straightforward.

**Weaknesses:**

1. The paper does not contain novel approaches. Typically, the authors only use transfer learning to extract image features and adopt LSTM / Transformers for video classification. The contribution is limited.
2. As mentioned in the paper, the gun is usually a small object in real-world videos. However, frame-level feature extraction is not an efficient way to get fine-grained-level features of guns.
3. In Table 4, it seems 100% accuracy has been achieved. I do not think it is a challenging task as the authors mentioned.

**Questions:**

1. There are some missing details in the experiments. I did not find how the dataset is divided into training set and testing set. The authors should make it clear.
2. What is the resolution of the real-world videos? I am curious whether the images contain sufficient features for guns.
3. Is the training shown in Figure 2 in an end-to-end manner or in multiple stages?
4. How to reduce the domain gap between gun images and real-world gun videos?

---

### Official Review · Reviewer_4hs6 · 2024-11-03

**Soundness:** 2
**Presentation:** 3
**Contribution:** 2
**Rating:** 5
**Confidence:** 4

**Summary:**

This paper addresses the limitations of existing video classification methods in gun detection by proposing to combine image-augmented training and temporal modeling. Experimental results demonstrate performance improvements on both synthetic and real-world datasets, particularly in firearm action recognition and the UCF crime dataset. The paper also highlights future research directions, including dataset diversity, model complexity, and ethical considerations.

**Strengths:**

1. I think gun detection is an important topic. However, the academia seems underrated this task and there are not many research on this direction. Gun detection has big potential to make our world safer. However, the exsiting methods have too many false positives to avoid its wide applications. Thus, I am glad to see that this work tries to contribute this important research direction.
2. The paper studies a valuable gun detection problem and provides a detailed motivation explanation.
3. The results show the improvements of the proposed methods.

**Weaknesses:**

1. The method proposed in the paper is a simple combination of existing techniques, such as common data augmentations like flipping and rotating.
2. The application of data augmentations is one of the core ideas of this paper, but it lacks specific details, such as how to select and adjust data augmentations.
3. The paper mentions using GRU and Transformer for temporal modeling, but it does not provide detailed descriptions of the configurations and optimization processes for these two models.
4. The paper implements a combined model of VGG and Transformer, but does not elaborate on how these models are integrated. For example, does it use a simple concatenation or parallel approach?
5. More importantly, gun detection typically requires a quick response in real-time monitoring systems, yet the paper does not address the model's real-time performance in practical applications.

**Questions:**

The optimal model combination (MobileNet + Transformer) has high complexity and computational resource requirements. Is this feasible for resource-limited practical applications, such as embedded devices and mobile devices?

---

### Official Review · Reviewer_ZF1V · 2024-11-04

**Soundness:** 2
**Presentation:** 2
**Contribution:** 2
**Rating:** 3
**Confidence:** 5

**Summary:**

This paper focuses on the challenging task of detecting tiny objects, specifically guns, in video data. Recognizing that current video analysis methods struggle with detecting guns due to the limited availability of labeled data and the complexity of identifying small objects in real-world scenarios, the authors propose a new method with three main contributions. They introduce a two-stage detection methodology that combines image-augmented training to improve spatial feature extraction with temporal modeling to capture sequential information. They validate the method on a synthetic firearms action recognition dataset and discuss key challenges and potential future research directions in tiny object detection.

**Strengths:**

1. The empirical study highlights the limitations of existing video classification methods, providing a solid foundation for the proposed approach. Additionally, the two-stage methodology enables the model to capture both spatial features of guns and temporal dependencies across frames, contributing to its effectiveness.
2. Given the real-world importance of detecting firearms in video data, the paper has notable significance. By addressing limitations in current methods and proposing a targeted approach, it contributes meaningful insights and methods that could inform future research for high-stakes applications.

**Weaknesses:**

1. Although the abstract highlights the challenges of limited labels and tiny object detection, the contributions do not directly address these points. Establishing a clearer link between the identified challenges and the proposed solutions would strengthen the study’s coherence and impact.
2. This paper does not clearly specify whether the focus is on object detection or action recognition, which may confuse readers regarding the study’s objective. A clearer explanation of the task would help define the scope and purpose of the proposed method.
3. The contributions are presented in an abstract manner, lacking concrete results or specific findings to support them. Adding more details about the outcomes or unique aspects of the contributions would make the study’s significance more evident.
4. The motivation section is overly lengthy and could be more concise. Streamlining this part would improve readability and make the core motivations clearer.
5. Section 4.1 contains excessive background details, including algorithmic descriptions of existing methods, which may not be necessary. Providing only the essential comparative results would make this section more concise and focused.
6. The proposed methodology in Section 4.2 is fairly basic, with limited technical depth, few formulas, and minimal emphasis on handling label scarcity or tiny object detection. Expanding this section to include more detailed techniques targeting these specific issues would add depth to the work.
7. Table 4 presents results on a dataset where all methods achieve 100% accuracy, making it an unnecessary addition. Removing or replacing it with a more challenging dataset would make the evaluation more meaningful.
8. The study does not compare against recent state-of-the-art methods and its variants from ablation studies, which limits the relevance and verification of the results. Including newer comparison methods would provide a stronger benchmark for the proposed approach.
9. The paper lacks qualitative visualizations, such as feature maps, example detections, or interpretability-focused analyses. Adding these would provide deeper insights into the method’s strengths and its effectiveness in handling complex detection scenarios.

**It is unfortunate that the authors, apart from a few commitments to revisions, have largely not addressed my concerns directly.**

**Questions:**

1. Is the study focused on object detection or action recognition? Please clarify to avoid confusion.
2. How do the proposed contributions specifically address label scarcity and tiny object detection?
3. Could you specify the novel aspects of your two-stage methodology compared to existing approaches?
4.The methodology in Section 4.2 appears basic. Could you provide more technical details, especially on handling tiny objects?
5. Comparisons are mostly with older methods. Could you add more recent state-of-the-art baselines?
6. Could you include qualitative visualizations, such as feature maps or example detections, to better demonstrate the model's performance?

---

> ### Comment · Reviewer_ZF1V · 2024-12-02
>
> Since the authors did not engage in the rebuttal process, I am inclined to maintain a somewhat negative rating.

---

### Meta-Review · Area_Chair_RpBm · 2024-12-18

**Metareview:**

This paper was reviewed by four experts in the field. The final ratings are 3,5,3,3. While reviewers agree gun detection is a useful and important problem, they raised many issues, such as novelty, motivation, insufficient experiments, etc. The authors did not engage in the rebuttal. There is no ground to overturn reviewers' recommendation.

**Additional Comments On Reviewer Discussion:**

Authors did not engage in the rebuttal period.

---

### Decision · Program_Chairs · 2025-01-22

Reject